# Snatching Sundews—Analysis of Tentacle Movement in Two Species of *Drosera* in Terms of Response Rate, Response Time, and Speed of Movement

**DOI:** 10.3390/plants11233212

**Published:** 2022-11-23

**Authors:** Caroline Ivesic, Wolfram Adlassnig, Marianne Koller-Peroutka, Linda Kress, Ingeborg Lang

**Affiliations:** 1Functional and Evolutionary Ecology, Faculty of Life Sciences, University of Vienna, Djerassiplatz 1, 1030 Vienna, Austria; 2Core Facility Cell Imaging and Ultrastructure Research, Faculty of Life Sciences, University of Vienna, Djerassiplatz 1, 1030 Vienna, Austria

**Keywords:** carnivorous plants, *D. allantostigma*, *D. rotundifolia*, tentacle types, snap-tentacles, movement dynamics, stimulus types

## Abstract

*Drosera*, Droseraceae, catch prey with sticky tentacles. Both Australian *Drosera allantostigma* and widespread *D. rotundifolia* show three types of anatomically different tentacles: short, peripheral, and snap-tentacles. The latter two are capable of fast movement. This motion was analysed after mechanical, chemical, and electrical stimulation with respect to response rate, response time, and angular velocity of bending. Compared to *D. rotundifolia*, *D. allantostigma* responds more frequently and faster; the tentacles bend with higher angular velocity. Snap-tentacles have a lower response rate, shorter response time, and faster angular velocity. The response rates for chemical and electrical stimuli are similar, and higher than the rates for mechanical stimulus. The response time is not dependent on stimulus type. The higher motility in *D. allantostigma* indicates increased dependence on mechanical prey capture, and a reduced role of adhesive mucilage. The same tentacle types are present in both species and show similar motility patterns. The lower response rate of snap-tentacles might be a safety measure against accidental triggering, since the motion of snap-tentacles is irreversible and tissue destructive. Furthermore, tentacles seem to discern stimuli and respond specifically. The established model of stereotypical tentacle movement may not fully explain these observations.

## 1. Introduction

In carnivorous plants, prey insects are caught by metamorphous leaves with various forms and characteristics. The attributes of trap-leaves enable classification by trapping mechanism: snap-traps, eel traps, suction traps, pitcher traps, and adhesive traps [1,2,3]. Adhesive trap-leaves in *Drosera* (Droseraceae) are covered by emergences, the so-called tentacles. These tentacles are multicellular stalked glands that produce sticky mucilage to catch insects and seal their stigmata, finally causing death by suffocation [1,3,4,5].

The first systematic assays of these leaves were performed by Charles Darwin (1875). He triggered motion by touch and various chemicals and observed movement of non-stimulated tentacles close to stimulated ones [6]. Since then, various stimuli triggering tentacle movement have been tested. Recent studies have shown that movements in *Drosera* are triggered by jasmonate signaling and action potentials, superficially comparable to action potentials in animal neurons [7,8,9,10]. Although *Drosera* includes more than 200 species [5,11,12] only a few are well investigated, especially the South African *D. capensis* [8,13,14,15,16,17].

In *D. capensis*, tentacles are quite uniform, whereas in other *Drosera* species, up to three morphologically different types can be distinguished. This study uses a modified version of the nomenclature by Poppinga et al., 2013 [18]. In *D. allantostigma* and *D. rotundifolia*, all three types can be discerned: snap-tentacles and peripheral tentacles at the leaf margin, and short tentacles on the leaf lamina. Peripheral tentacles have radially, and snap-tentacles bilaterally symmetrical, glandular heads (Figure 1). The term “snap tentacle“ was originally coined for morphologically aberrant and highly motile tentacles in some Australian sundews [19,20]. However, this tentacle type is not restricted to Australian species, and has even been described for various species under different names long ago. For instance, Fenner described and depicted marginal, bilaterally symmetrical tentacles (recently known as snap-tentacles) in 1904 in European *D. rotundifolia* specimens [21].

Snap-tentacles have only recently drawn attention to the scientific community due to their fast catapulting movement in the Australian *D. glanduligera* (with 75 ms from first mechanical touch to complete bending of the tentacle) [18,19,22,23]. Thus, functional specialization of tentacle types could be based in a differentiated response to stimuli. Snap-tentacles may move faster but would be less involved in subsequent prey degradation and utilization. However, it remains unknown if these types differ generally in their physiology.

Though the principal mechanism of tentacle movement has been clarified, the movement in three-dimensional space has not yet been characterized in quantitative terms. Furthermore, it is unknown if movement can be triggered directly by an adequate electrical signal as in the related *Dionaea muscipula* [6]. In addition, the relation between movement attributes such as, e.g., response frequencies and stimulus types, remains to be explored.

## 2. Results

### 2.1. Descriptive Statistics

Tentacles reacted more frequently to chemical and electrical stimuli than to mechanical stimulation (Table 1). In the course of this study, a total number of 1441 tentacles were exposed to electrical, chemical, or mechanical stimuli. Of these, 718 reacted, coming down to a total response rate (RR) of 49.8%. In *D. allantostigma*, RR was significantly higher (chi-squared test, *p*-value < 0.001), with 57.6% compared to *D. rotundifolia* with 45.8%. In spite of that, the highest RR (78.0%) was found for peripheral tentacles in *D. rotundifolia.* The shortest (i.e., the fastest) response time (RT) was observed in the peripheral tentacles of *D. allantostigma* when stimulated electrically. This prompt response of 9 s was countered by the prolonged RT in snap-tentacles of *D. rotundifolia* (51 s) under the same stimulus. The highest angular velocity (AV), 7.38°/s was found in mechanically stimulated snap-tentacles of *D. allantostigma*. The slowest AV with 0.65°/s was detected in mechanically stimulated snap-tentacles of *D. rotundifolia* (Table 1).

Overall response rates, response times, and angular velocities showed a high variability, and the respective effect of the different stimuli was by no means obvious (Table 1). Thus, linear (for RT and AV) and logistic regressions (RR) were used, first with species, tentacle types (Table 2), and in extension with stimulus types as independent variables (Table 3). All the models explained only a small part of the total variability (R² ≤ 0.27) but were significant (F < 0.05). Thereby, the models for response rate and angular velocity explained more of the observed variability than the models for response time (Table 2 and Table 3).

### 2.2. Tentacle Type Specific Responses

The snap-tentacles of both species show adhesion of fish food in the observed video material. Furthermore, mucicarmine staining to test for mucilage presence on tentacle heads revealed the presence of mucus around the glandular tissue of snap-tentacles (Figure 2). Table 2 shows that tentacle types do not differ in any movement parameters when triggered electrically. However, a significant difference between tentacle types can be found for RR and AV when triggered chemically and mechanically. In general, snap-tentacles react less frequently to both stimuli than peripheral tentacles (Table 1 and Table 2).

Comparing AV between tentacle types in *D. allantostigma*, the snap-tentacles answer with higher AV than peripheral tentacles when stimulated mechanically or chemically (Table 1). AV in *D. rotundifolia* also depends on the tentacle type; however, here snap-tentacles exhibit faster motion when triggered chemically, but not mechanically or electrically (Table 1). In terms of RR and AV, the tentacle types significantly differ from one another when triggered chemically and mechanically (Table 2).

The RT is not influenced by tentacle type (*p*-values > 0.05). These results suggest that within one stimulus type, signal transduction in morphologically different tentacles functions in the same way, making the RT invariable (Table 2).

Without regard to tentacle types, the models delineate that there is a clear difference in the RR, RT, and AV between species (Table 2 and Table 3).

### 2.3. Species-Specific Division of Labour

*D. rotundifolia* has lower RR (Table 3). *D. allantostigma* has shorter response times than *D. rotundifolia* regarding mechanical (*p*-value < 0.05) and electrical (*p*-value < 0.001) stimuli, but the opposite is true concerning chemical stimuli (Table 2). When stimulated chemically, *D. rotundifolia* responds faster (Table 2). In spite of *D. rotundifolia* responding faster to chemical triggering, the models comprising all independent variables show that *D. allantostigma* responds, in general, faster in terms of RT (Table 3). Considering AV, *D allantostigma* exhibits substantially faster motion than *D. rotundifolia* (Table 3).

## 3. Discussion

The motion of tentacles in *Drosera* is triggered by action potentials, which, in turn, are triggered by jasmonate signaling [7,8,24,25]. The jasmonate release is highly dependent on stimulus type, with chemical stimuli producing elevated jasmonate pools as compared to mechanically stimulated ones [26]. Elevated jasmonate pools provoke continuous signal transduction, subsequently triggering action potentials [24]. *D. muscipula* is able to ’count’ stimuli [24,27]. This snap-trap species is very closely related to *Drosera* and the triggering of trap-leaves functions in a similar way to the triggering of tentacles [18,28,29].

Chemical triggering might increase jasmonate presence and cause heightened response frequencies in the observed tentacles (Table 1), following the rule of more jasmonate—more intense action potentials—more response. Another explanation could be that the continuity of the signal also plays a role, following the same rule as in *D. muscipula*: More jasmonate—more continuity of signal—higher signal count—more response [24]. These two theories are not mutually exclusive and could both justify the results. Both theories reasonably imply that the tentacles are able to discern a mechanical stimulus from a chemical or electrical one.

In extension, similarly high response rates were gathered from electrical stimulation (Table 1). Electric pulses of 16 kV were required to trigger any reaction, compared to action potentials of < 100 mV found in *D. capensis* [7]. The need for this excessive voltage might be explained by the extremely low electric conductivity of the air separating the electrode and the tentacle and the thick cuticle of the tentacle. By overruling the processes generating action potentials in the natural habitat, every tentacle that is capable of movement responds to a strong electrical stimulus. Thus, the “decision“ made by the tentacles, whether an action potential is generated and motion is initiated, is overridden. The detailed models using electrical stimulation as an independent variable also reflect this rule. Different tentacle types detect, respond, and react similarly to an adequate electrical signal (Table 2).

The snap-tentacles of pygmy sundews are described as glueless [18,22]. Having lost the capability of catching prey via sticky mucilage, these tentacles have adapted to prey retention by mechanical means [30]. Snap-tentacles snap prey onto peripheral and short glue tentacles in the leaf middle in order to enable dissolving and uptake [18,22,30]. Even though the movement times (*D. allantostigma*: 20 s and *D. rotundifolia*: 62 s from movement start to movement end) do not come close to *D. glanduligera* (75 ms until full bending of the tentacle) [30], *D. allantostigma* seems to follow a similar evolutionary trend towards mechanical prey capture and retention instead of glue tentacle mediated prey capture by adhesion forces. This response to prey makes specialized tentacles an essential prerequisite.

The response time depends on species and tentacle type but not on stimulus type (Table 3). This finding suggests that the signal transduction itself and the determination of a response are not at all relying on the type of stimulus.

In *D. glanduligera*, the fast catapulting motion is highly tissue destructive and irreversible at the level of hinge zones [18,30]. In order to protect themselves from false triggering, the snap-tentacles would respond less to chemical and mechanical stimulation and only move when the signal is adequate, which reflects the RR`s offered in this study (Table 1).

Based on theoretical calculations by Skotheim and Mahadevan [31], Poppinga et al. hypothesized that the movement of snap-tentacles is linked to hinge zone elasticity and hydraulic pressure [30]. The angular velocity differs significantly between the species that are tested. This could be due to differences with regard to the hydrostatic pressure build up at hinge zones, or to differences of the elasticity of the tissue. *D. allantostigma*´s tentacles are smaller, enabling faster volume flows compared to the hinge zone of *D. rotundifolia*, resulting in overall faster movement. These observations indicate a strong specialization of snap-tentacles toward mechanically triggered prey retention.

*D. rotundifolia*´s tentacle heads are significantly bigger than the ones of *D. allantostigma* [32]. The fast response of both tentacle types in *D. rotundifolia* by chemical stimulation could be due to considerably enlarged surface area composed of glandular epidermis cells. Therefore, more information and more signals can be transmitted through the cells, resulting in a more pronounced trigger of jasmonate signaling. This subsequently starts the formation of action potentials that trigger motion. *D. rotundifolia* shows specialization for chemical triggering, but there are only minor differences in movement behaviors of the tentacle types. Additionally, the adhesion force in snap-tentacles is due to mucus presence (Figure 2), which shows that this tentacle type is sort of an all-rounder for prey capture.

## 4. Materials and Methods

### 4.1. Plant Material

Specimens of *D. allantostigma* and *D. rotundifolia* were grown at the greenhouses of the Bundesgärten Schönbrunn and then transferred to and kept in the greenhouse of the former biocentre (Althanstraße 14, 1090 Vienna) of the University of Vienna for the experimental series. Plants were kept under high sunlight exposition, warm temperatures (23 °C), and high humidity. All macroscopical investigation and video documentation was performed at the Core Facility Cell Imaging and Ultrastructure Research, University of Vienna—member of the Vienna Life-Science Instruments (VLSI).

### 4.2. Mucilage Staining

Mucilage staining was performed to localize mucus production around gland heads of all tentacle types. A 2% solution of mucicarmine from powder (Riedel-de Haën, product number: 32787) diluted in distilled water was produced. Trap leaves were cut off at the petiole and transferred to a microscopic slide and the staining solution was dripped onto the leaves. Excess solution was absorbed by filter paper. Either whole leaves (*D. allantostigma*) or sections of the periphery (*D. rotundifolia*) were covered with a droplet of water and with a cover slip. For microscopic imaging, a Nikon ECLIPSE Ni microscope connected to a Nikon DS-Ri2 camera in combination with the NIS-Elements software was applied.

### 4.3. Stimulation Experiment

Overall, 56 trap-leaves of 21 rosettes of different *D. allantostigma* individuals were exposed to chemical (16 trap-leaves), mechanical (22 trap-leaves), and electrical stimuli (18 trap-leaves). In *D. rotundifolia*, 89 trap-leaves of 16 rosettes of different individuals were exposed to stimulation (19 trap-leaves to electrical, 36 to chemical, and 34 to mechanical). All stimuli were applied by a modified micromanipulation tool by LEITZ so that individual tentacles could be stimulated without touching others. Two tentacle types were stimulated: snap-tentacles and peripheral tentacles.

For mechanical stimulation, tentacles were touched with a chemically inert platinum needle, held by the micromanipulator, at the glandular head as well as on the stalk. For chemical stimulation, fish food (‘sera San’^®^) was ground with a pestle and mortar and wetted with distilled water. Fish food was chosen because preliminary experiments had shown that tentacles show a stronger reaction to heterogeneous mixtures of nutrients than to isolated compounds; fish food is rich in protein and has a reproducible composition (48.8% crude protein, 8.4% of crude fat, 11.4% crude ash, 5.0% water, and 3.2% crude fiber) [33]. Single flakes (≈0.5 mm) were attached to a glass pipette tip. The pipette with fish food was brought into contact with an individual tentacle head by the micromanipulator. For electrical stimulation, a thin tungsten wire was positioned by the micromanipulator in a distance of 1 mm to tentacle heads (Appendix A). Electric voltage was generated by a piezo igniter with a range of 12–16 kV adding two resistances coherent to 8 Ω each. Preliminary experiments had shown no tentacle movement after stimulation by a direct current generator with an electric potential ≤ 17 kV.

Tentacle movement was documented using a Wild Photomacroscope M400 and a Nikon 1J1 camera. Videos were analyzed with regard to response rate, response time, and specific angles. Raw data were processed as follows:Response rate (RR): For each species, tentacle type, and stimulus, the number of tentacles responding within 5–7 min and non-responding tentacles were counted.Response time (RT) was defined as the times [s] passing between the first contact with the stimulus (t_0_) and the start of the motion (t_1_) (RT = t_1_ − t_0_). In case of electrical stimulation, multiple stimuli were applied; t_0_ was the time of the first triggering of the piezo igniter.Angular velocity (AV) was calculated by applying the trigonometric model following the description in Section 4.4.

### 4.4. The Trigonometric Model for Approximation of Angular Velocity (AV)

The relatively stiff tentacle stalk bends around a joint, the so-called hinge zone [24]. Since the tentacle stalks vary in length, the absolute speed of movement [mm/s] provides no meaningful information. A superior measure of the movement is given by the angle at the hinge zone. The videos made for measurements of angular velocity delivered a two-dimensional (2D) projection of a movement in three dimensions (3D). The movement towards the camera from t_1_ to t_2_ was measured by the perspective shortening of the tentacle (Figure 3).

The component of movement parallel to the focal plane was measured directly (length b). Figure 4 shows how these measurements were merged in order to determine the angular movement.

φ is the real angle of movement and x is equivalent to the real distance in the 3D space that the tentacle tip surpassed. To find the real tentacle tip movement x, first h had to be calculated, which was the theoretical height (the movement towards the camera) the tentacle had reached. From Pythagoras theorem, we can apply these two parts:(1)$$h=\sqrt{a_{1}{}^{2}-a_{2}{}^{2}}$$
(2)$$x=\sqrt{h^{2}+b^{2}}$$The angle was then calculated according to the formulas:(3)$$\sin\left(\frac{\varphi }{2}\right)=\frac{x/2}{a_{1}}$$
(4)$$\varphi =2\times \;\arcsin\left(\frac{x/2}{a_{1}}\right)$$The angular velocity (AV) was defined as φ /(t_2_ − t_1_), measured in °/s.
(5)$$\mathrm{AV}=\frac{\varphi }{\left(t_{2}-t_{1}\right)}$$

The different motion parameters were used to compare the reaction to different stimuli as well as to elucidate differences between motion dynamics in the two tentacle types and species. The required sample size was determined by power estimation based on preliminary experiments in 109 tentacles. The study was powered to detect a significant difference (*p*-value < 0.05) between subgroups of all motion parameters with a probability of 80%. The three aforementioned parameters were assigned into subgroups, itemized by species (*D. allantostigma*/*D. rotundifolia*), tentacle type (snap-tentacles/peripheral tentacles), and stimulus type (mechanical/chemical/electrical). Response rates in %, medians in [s] for response time, and medians in °/s for angular velocities were used for descriptive statistics. Chi-squared test was used to detect differences in response rates. Response time and angular velocity were first checked for normal distribution and homogeneity of variance via the Bartlett’s test. As normal distribution and homogeneity of variance were not given, the Kruskal–Wallis test was used to compare the subgroups. If the Kruskal–Wallis test results revealed significant differences between groups, Dunn’s test was used to detect where the exact differences came from.

High variability strongly influenced the results and hindered the exposure of any clear trend. Hence, regression analyses were applied to model similarities and differences between the observed subgroups. Here, the movement sequence parameters were used as dependent variables and were regressed against species, tentacle types, and stimulus types. Statistics were programmed and calculated in Stata^®^ V14.

## 5. Conclusions

Subdividing the movement sequence of tentacles in *Drosera* into specific parameters is necessary, and facilitates statistic numeric analysis. From the results we can conclude:The type of stimulus plays a significant role in response rates and angular velocities. However, it does not influence the response time. Chemical and electrical stimulation produce virtually identical responses. In comparison, mechanically stimulated tentacles react less frequently, with the same response time and a higher angular velocity.Movement sequence parameters are strongly dependent on species. In *D. allantostigma* tentacles react more frequently, with a shorter response time and higher angular velocity compared to *D. rotundifolia*. Snap-tentacles respond less frequently but with shorter response times and higher angular velocities compared to peripheral tentacles.Tentacle responses are highly species-specific, with *D. allantostigma* exhibiting a specialization to mechanical stimulus, displayed by the highest angular velocities observed. This is especially pronounced in the movement sequence parameters of snap-tentacles in comparison to peripheral tentacles. *D. rotundifolia* is specialized in chemical triggers, and this type of stimulus leads to the highest response rates and angular velocities within this species.Finally, this study shows that behavior of snap-tentacles and peripheral tentacles varies between the species that were observed. *D. allantostigma* has snap-tentacles specialized in mechanic prey capture and retention, whereas snap-tentacles in *D. rotundifolia* behave similarly to peripheral tentacles, with adhesion and mechanical prey retention.

## Figures and Tables

**Figure 1 plants-11-03212-f001:**
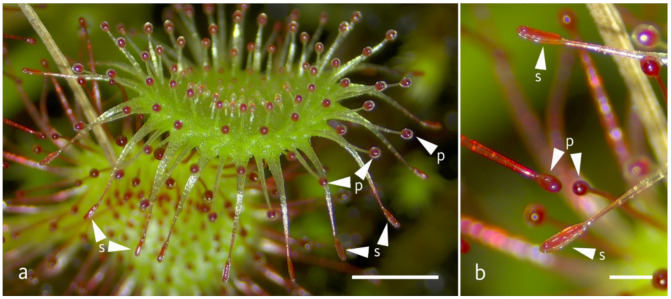
Figure of a *D. rotundifolia* (**a**) trap-leaf (scale bar = 2 mm) with peripheral and snap-tentacles and (**b**) detail of trap with peripheral and snap-tentacles (scale bar = 0.5 mm); examples for peripheral tentacles are signed with arrow heads (p) and for snap-tentacles with arrow heads (s).

**Figure 2 plants-11-03212-f002:**
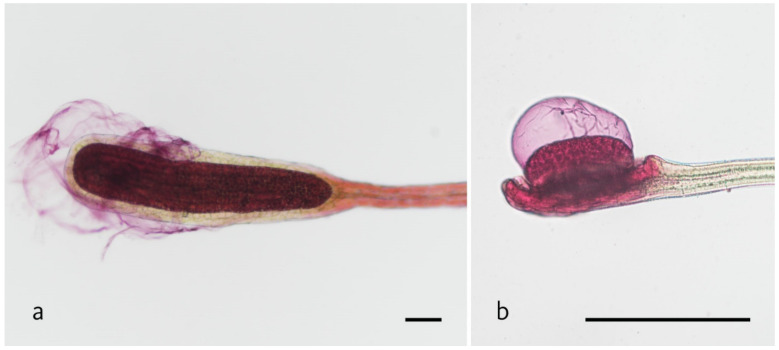
Depicts the mucus layer on the head of the snap tentacles with mucicarmine staining of (**a**) *D. rotundifolia* and (**b**) *D. allantostigma.* Scale bar indicates 150 μm.

**Figure 3 plants-11-03212-f003:**
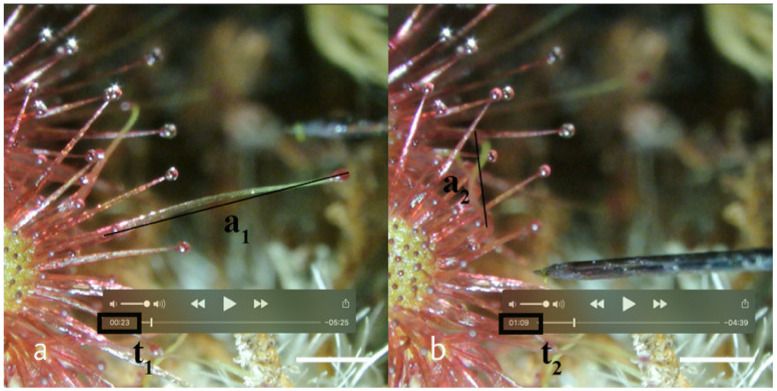
Example of a snap-tentacle bending in *D. allantostigma*. Measurements made to calculate the movement parameter angular velocity. a_1_ is the measured on-screen tentacle length at the start of movement t_1_ (**a**), a_2_ is the measured on-screen tentacle length at the end of movement t_2_ (**b**). A detailed video can be found in the Appendix A. Scale indicates 1 mm.

**Figure 4 plants-11-03212-f004:**
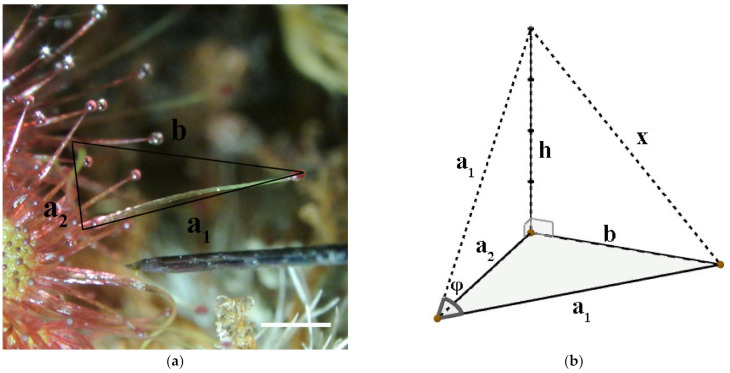
Determination of angular movement. (**a**) Merged image of tentacle movement at t_1_ and t_2_. a_1_ = apparent tentacle length at t_1_/a_2_ = apparent tentacle length at t_2_/b = on-screen movement of the tentacle head from t_1_ to t_2_. Scale indicates 1 mm. (**b**) 3D reconstruction of the movement. The plane a_1_ a_2_ b is parallel to the focal plane of the macroscope. The tentacle head moves along x by the angle φ.

**Table 1 plants-11-03212-t001:** Summary table of response rates (RR) with 95% confidence intervals (LL = lower confidence level, UL = upper confidence level)/median response times (RT)/median angular velocities (AV). RT and AV with 1st Quartile (Q1) and 3rd Quartile (Q3).

Stimulus Type	Tentacle Type	Species	RR			RT [s]			AV [°/s]		
			%	LL	UL	M	Q1	Q3	M	Q1	Q3
Mechanical Stimulation	Snap- tentacles	*D. allantostigma*	31.2%	20.9%	41.5%	17.5	10.0	36.2	7.38	1.0	15.8
		*D. rotundifolia*	15.0%	8.6%	21.4%	38.0	16.0	65.0	0.65	0.3	1.2
	Peripheral tentacles	*D. allantostigma*	56.8%	45.5%	68.1%	15.0	10.0	33.0	4.08	1.9	7.3
		*D. rotundifolia*	19.7%	14.9%	24.5%	26.0	18.0	41.5	0.98	0.6	1.2
Chemical Stimulation	Snap- tentacles	*D. allantostigma*	49.1%	39.9%	58.3%	35.5	26.3	57.5	3.41	1.1	5.1
		*D. rotundifolia*	44.0%	32.8%	55.2%	19.0	10.0	48.0	1.24	0.4	2.3
	Peripheral tentacles	*D. allantostigma*	71.2%	60.8%	81.6%	28.0	17.0	58.3	1.25	0.5	2.9
		*D. rotundifolia*	78.0%	71.7%	84.3%	25.0	16.5	35.5	1.11	0.7	1.8
Electrical Stimulation	Snap- tentacles	*D. allantostigma*	71.4%	54.7%	88.1%	31.5	15.8	40.5	3.52	1.8	5.3
		*D. rotundifolia*	54.5%	39.8%	69.2%	51.0	17.8	75.8	0.84	0.4	1.4
	Peripheral tentacles	*D. allantostigma*	70.9%	63.0%	78.8%	9.0	6.0	36.0	3.85	2.1	6.1
		*D. rotundifolia*	62.8%	57.2%	68.4%	39.0	17.0	73.0	0.76	0.5	1.4

**Table 2 plants-11-03212-t002:** Regression models with species and tentacle types as independent variables, subgrouped by stimulus types. The species variable assumes the value 0 for *D. rotundifolia* and 1 for *D. allantostigma*. The tentacle type variable assumes 0 for peripheral tentacles and 1 for snap-tentacles. For RR coefficients, < 1 indicate reduced rates. For RT and AV, coefficients < 0 indicate smaller values in the species/tentacle types coded with 1.

Dependent Variable	Independent Variable	n	R²	Prob > F	Species		Tentacle Type	
					*p*-Values	Coefficient	*p*-Values	Coefficient
RR	Electrical Stimulation	481	0.01	0.095	0.042	1.54	0.419	0.81
	Chemical Stimulation	430	0.07	<0.001	0.806	0.95	<0.001	0.29
	Mechanical Stimulation	530	0.07	<0.001	<0.001	4.06	0.004	0.51
RT	Electrical Stimulation	310	0.11	<0.001	<0.001	−35.29	0.078	14.03
	Chemical Stimulation	266	0.04	0.004	0.002	24.77	0.852	1.53
	Mechanical Stimulation	135	0.05	0.034	0.012	−23.69	0.358	9.22
AV	Electrical Stimulation	311	0.27	<0.001	<0.001	3.16	0.998	0.00
	Chemical Stimulation	272	0.12	<0.001	<0.001	1.43	0.026	0.73
	Mechanical Stimulation	134	0.19	<0.001	<0.001	7.53	0.014	4.45

**Table 3 plants-11-03212-t003:** Models including all independent variables: Species, Tentacle types, Stimulus types. The variable species assumes *D. rotundifolia* as 0. Peripheral tentacles are assumed the value 0. Both chemical and mechanical stimulation are compared to electrical stimulus (assumed 0).

Dependent Variable	n	R²	Prob > F	Species		Tentacle Type		Stimulus Chemical		Stimulus Mechanical	
				*p*-Values	corr. coef	*p*-Values	corr. coef	*p*-Values	corr. coef	*p*-Values	corr. coef
RR	1441	0.13	<0.001	<0.001	1.73	<0.001	0.45	0.451	1.12	<0.001	0.21
RT	711	0.02	0.012	0.007	−11.65	0.018	11.89	0.314	−4.72	0.125	−8.84
AV	717	0.15	<0.001	<0.001	3.35	0.008	1.16	0.087	−0.69	<0.001	1.85

## Data Availability

Raw data will be made available upon request.

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
