# Peer review of "Snatching Sundews—Analysis of Tentacle Movement in Two Species of Drosera in Terms of Response Rate, Response Time, and Speed of Movement"

_plants, 2022, doi:10.3390/plants11233212_

Round 1
Reviewer 1 Report
This is a very interesting and sound study on carnivorous sundew tentacle behaviour, which I am happy to recommend for publication after the following minor points have been solved. My sincere congratulations to the authors!
Please check for typos, e.g., missing blank spaces or dots in species names (D.allatostigma).
Abstract: “catch prey by sticky tentacles” should probably be “catch prey with”
Introduction: “Australian pygmy D. glanduligera”
D. glanduligera is not a pygmy sundew, it belongs to another section!
Section 2.2: “This could be due to the response in snap-tentacles being highly tissue destructive and the plant trying to protect itself from it by only initiating the movement if the stimulus is adequate”.
This statement also appears elsewhere, for example in the abstract. The observation that the base of snap tentacles can easily rupture was only made for D. glanduligera so far. What is the significance for the species studied here? Was the same observation made? If yes, please show data – otherwise, redact this passage.
Section 2.2: “D. allantostigma seems to follow a similar evolutionary trend towards mechanical instead of trap-mediated prey capture”.
I must admit that I was a bit lost here. What do the authors mean with the differentiation between mechanical and trap-mediated prey capture? Please explain further.
Discussion: “Poppinga et al. hypothesized that the movement is linked to hinge zone elasticity and hydraulic pressure [24]” […] “D. allantostigma´s tentacles are smaller, the elastic energy at hinge zones is thereby smaller than that stored in the hinge zone cells of D. rotundifolia. However due to the tiny size of the tentacles this smaller elastic energy in hinge zones of tentacles in D. allantostigma has an even bigger effect on the movement parameter of angular velocity, resulting in overall faster movement.”
Poppinga et al. were not able to directly evaluate the effect of mechanical actuation (i.e., the release of prestress) and hydraulic actuation (i.e., turgor changes). They argued that the theoretical timescale for hydraulic actuation lies way below the actually measured speed. This, and in addition the fact that no obvious material for storing elastic prestress was observed, led them to argue that hydraulic actuation alone is likely to drive fast snap-tentacle motion in D. glanduligera. This does indeed also suit well to the observations made in this study, because the smaller tentacles of D. allantostigma could be actuated faster than those of D. glanduligera (due to the shorter poroelastic time, calculated after Skotheim & Mahadevan (2005) Science). Therefore, I recommend to alter this section accordingly.
Discussion: “Additionally, the adhesion force in snap-tentacles is due to mucus presence, which shows that this tentacle type is sort of an all-rounder for prey capture.”
This mucus presence was also mentioned before – can the authors provide evidence for this? Probably I missed it or figures are missing in the ms?
Materials and Methods: The authors do not indicate how many tentacles were used from how many plants, i.e., were there groups of tentacles from identical plants used? The authors should discuss this in context of their statistical analyses.
Conclusion: “Finally, this study shows that snap-tentacles and peripheral tentacles behaviour is highly species specific.”
I recommend being careful with such dramatic statements, since only two species were investigated comparatively.
Reviewer 2 Report
I have always loved Drosera and so was excited to read and review this manuscript. The authors have address an interesting question - how do two species of Drosera compare to one another, and to other species studied previously, in the response of their different types of trichomes to stimuli? I have some unanswered questions that could be clarified in the Methods section and in the presentation of data: How many plants of each species were studied? How many leaves? Even if it is only one, that should be clearly stated. And in the data tables, a single number is given, yet the authors state there is a lot of variability in the responses and measures. That should be evident in the data presented, but it is not.
I enjoyed the detailed mathematical explanations, but am not sure what the modeling accomplished, maybe that could be explained more clearly.
I used the Comment tool in Adobe pdf to suggest changes and edits to wording used. The document is attached.
